# National Scale Spatial Variation in Artificial Light at Night

**Daniel T.C. Cox** [1,*], **Alejandro Sánchez de Miguel** [1], **Simon A. Dzurjak** [1], **Jonathan Bennie** [2] and **Kevin J. Gaston** [1]

1   Environment and Sustainability Institute, University of Exeter, Penryn, Cornwall TR10 9FE, UK;
    A.Sanchez-De-Miguel@exeter.ac.uk (A.S.d.M.); S.Dzurjak@exeter.ac.uk (S.A.D.);
    K.J.Gaston@exeter.ac.uk (K.J.G.)
2   Centre for Geography and Environmental Science, University of Exeter, Penryn, Cornwall TR10 9FE, UK;
    J.J.Bennie@exeter.ac.uk
*   Correspondence: d.t.c.cox@exeter.ac.uk; Tel.: +44 (0) 1326 259490

**Abstract:** The disruption to natural light regimes caused by outdoor artificial nighttime lighting has significant impacts on human health and the natural world. Artificial light at night takes two forms, light emissions and skyglow (caused by the scattering of light by water, dust and gas molecules in the atmosphere). Key to determining where the biological impacts from each form are likely to be experienced is understanding their spatial occurrence, and how this varies with other landscape factors. To examine this, we used data from the Visible Infrared Imaging Radiometer Suite (VIIRS) day/night band and the World Atlas of Artificial Night Sky Brightness, to determine covariation in (a) light emissions, and (b) skyglow, with human population density, landcover, protected areas and roads in Britain. We demonstrate that, although artificial light at night increases with human density, the amount of light per person decreases with increasing urbanization (with per capita median direct emissions three times greater in rural than urban populations, and per capita median skyglow eleven times greater). There was significant variation in artificial light at night within different landcover types, emphasizing that light pollution is not a solely urban issue. Further, half of English National Parks have higher levels of skyglow than light emissions, indicating their failure to buffer biodiversity from pressures that artificial lighting poses. The higher per capita emissions in rural than urban areas provide different challenges and opportunities for mitigating the negative human health and environmental impacts of light pollution.

**Keywords:** albedo; landcover; light emissions; light pollution; protected areas; skyglow; sky brightness; urbanization

## 1. Introduction

Outdoor artificial nighttime lighting, from streetlights and other sources, has increasingly been recognized as a significant anthropogenic pressure on the environment (e.g., [1–3]). The disruption it causes to natural light regimes has a wide range of impacts, in large part because of the central role that these regimes commonly play in determining the timings of biological activity [4]. Outdoor artificial nighttime lighting may contribute to a variety of chronic human health conditions ([5–7]; but see [8]), and the loss of views of celestial night skies has been argued to have a significant influence on people's sense of place [9], although, in both cases, empirical evidence is limited. The evidence is clearer with regard to other organisms, with impacts of outdoor artificial nighttime lighting on gene expression [10], budburst and flowering times [11,12], hormone cycles [13], activity patterns [14–19], foraging success [20,21], growth [22,23],

mortality risk [24,25], trophic interactions [26,27], population abundances [28,29], migration patterns [30–32], community structure [33–36] and ecosystem functioning and services [37–39].

Key to determining where the biological impacts of artificial nighttime lighting are likely to be experienced, is an understanding of the spatial occurrence and variation in that lighting, and particularly how this changes with other factors [3]. To date, from this perspective, this understanding has been rather limited. The most frequently studied correlates of spatial variation in artificial nighttime lighting have been levels of urbanization, human density and economic activity (e.g., [40–47]). In part, this focus has been because urban infrastructure is clearly a major contributor to artificial nighttime lighting, and in part because of interest in the potential to use remotely sensed data of such lighting to track or hindcast changes in urbanization and human density, where such information is not directly available.

A few studies, typically at either very large (e.g., global) or quite small scales (e.g., cities, streets or microhabitats), have examined how artificial nighttime lighting changes with ecosystem type, land cover or land use. At a global scale, it has been found to be most prevalent in temperate climate ecosystems, followed by Mediterranean and tropical/subtropical ones [48], and within urban areas to decline with levels of greenness and to increase with coverage by roads [44,49,50]. In Brazil the most heavily affected vegetation types are terrestrial coastal (restingas and mangroves), semideciduous seasonal forests and mixed ombrophilous forests [51]. Within individual cities or parts thereof, there have been attempts to determine the different sources of lighting (e.g., [50]). Research at more intermediate scales has been lacking.

A small number of studies have also examined the levels of artificial nighttime lighting within and around areas protected for biological conservation (global—[52–55]; China—[56]). The designation, establishment and maintenance of protected areas is widely considered key to buffering biodiversity from a diverse range of anthropogenic pressures [57,58]. However, whilst levels of artificial nighttime lighting tend, overall, to be lower in these areas, many are not escaping such influences [53,54,56], and lighting is particularly increasing in their vicinity, with implications for the lighting of horizons observed from within [55].

One can recognize two different forms of artificial light at night: light emissions that come directly from sources, and diffuse skyglow (brightening of the nighttime sky that results from artificial light emissions being scattered by water, dust and gas molecules in the atmosphere). Almost invariably, to date, studies of spatial variation in artificial nighttime lighting have focused on measures of light emissions (but see [59]).

In this paper, we determine how light emissions and skyglow vary with human density, land use, roads and protected areas for the first time at a national scale, using Britain as a case study. This makes an interesting exemplar, as it is an area in which the environmental impacts of artificial nighttime lighting have attracted a good deal of policy and management attention (e.g., [60–64]). We demonstrate that, although artificial light at night increases with human population density, the amount of light per person decreases with increasing urbanization. Overall, the amount of artificial light at night varies significantly across different land uses, and we demonstrate that light pollution is no longer solely an urban issue.

## 2. Materials and Methods

### 2.1. Study Area

This study focused on mainland Britain (51°51'N, 0°12'E), which occupies c.205,000 km$^2$ and, in mid-2015, held a population of c.63.3 million people [65]. All spatial manipulations were performed in Google Earth Engine [66] and R software for statistical computing version 3.5.2 [67] using the packages 'raster' (version 2.8.19; [68]), 'sf' (version 0.7.3; [69]) and 'spdep' (version 1.3.2; [70]). All data rasters were downloaded at a spatial resolution of 0.0083 degrees (c.1 km$^2$ at the equator), unless otherwise stated, and reprojected to the British National Grid (EPSG:27700). The resulting raster resolution at this latitude was 522 m by 927 m (c.500,000 m$^2$). To include only mainland Britain, we created a spatial

polygon of the coastal outline of England, Scotland and Wales, thereby excluding outlying islands. This layer was used in conjunction with the 'mask' function on each of the following seven rasters in turn; all pixels where the centroid fell outside of this polygon were designated NA.

*2.2. Data Sources*

2.2.1. Light Emissions

To arrive at a measure of light emissions, we took advantage of the Monthly Average Radiance Composite images (version 1), created from the Visible Infrared Imaging Radiometer Suite (VIIRS) day/night band (DNB; [71]). The DNB is sensitive to light in the range 500 to 900 nm, thereby covering the near-infra-red region beyond the range of the human eye, but excluding the blue and violet parts of the visible spectrum, for which no public data are currently available. The data are composited monthly, therefore, to obtain a single measure across years for which data were available (2012 to 2018), we calculated the median average radiance from August to November for each year. These months were chosen to avoid the confounding effect of reflectance from seasonal snowfall in winter and spring, increased leaf cover in spring and reduced levels of darkness in the middle of summer. We then calculated the mean radiance across years (mcd/m$^2$). A single averaged measure of light emissions does not account for growth through time in the radiance of light emissions in the UK [72]. However, here we were interested in spatial variation and, although related to growth, it is several orders of magnitude greater, and so temporal changes are not comparable.

To obtain as close a measure to light emissions as possible we corrected for albedo and skyglow. The inclusion of albedo would create potentially misleading and complicated models, where the albedo is subtracted from both artificial and natural readings, and no model currently exists that is able to do this. To correct for albedo, we used the MODIS MCD43B3.005 Albedo 16-day global 1 km product generated by the National Aeronautics and Space Administration (NASA). We selected three bands that fell within the spectrum of light to which the DNB is sensitive (BSA Band1, 620-670nm; BSA Band4, 545-565 nm; Albedo_BSA_nir, 858 nm). For each band, we generated a single layer, calculating the median albedo from August to November for each year from 2012–2018), before taking the mean across years. This is the same period as covered by the VIIRS day/night band above, the only difference being the albedo medians are for each day during this period, whilst VIIRS data is only collected on moonless nights. This approach produced the highest signal to noise ratio, and is therefore the most accurate [73]. We did not control for change in albedo due to vegetation during these months, because these changes are relatively small compared to factors such as the view angle of VIIRS, snow cover or acquisition time [74,75]. We calibrated the albedo on a desert region with examples of high and low albedo and where there is very little change in albedo across the year, therefore providing one of the most temporally stable and highest albedo gradients in the world (area = c.55 km * c.45 km; 24°90'N, 17°74'E), we then plotted each of the three resulting albedo layers against the DNB and calculated the linear regression, selecting the layer with the best fit (Albedo_BSA_nir, pR$^2$ = 0.72). For each pixel on the DNB layer we calculated the light emissions corrected for albedo (DNBac) as:

$$DNBac = DNB \; of \; pixel - (Albedo^* \times constant) \tag{1}$$

$$Albedo^* = (Albedo \; of \; pixel - Albedo \; minimum^+)/(Albedo \; maximum^+ - Albedo \; minimum^+) \tag{2}$$

where *constant* is the linear slope of albedo_BSA_nir plotted against DNB obtained from the region, with examples of high and low albedo [75], and $^+$ denotes that values were obtained from the same region.

Finally, we corrected for skyglow by subtracting the pixel value for skyglow obtained from Falchi et al. ([59]; see below) from the DNBac. Because the minimum artificial light at night is 0, any resulting negative values were allocated a value of 0.08, equivalent to the intensity of the natural sky brightness and the median light emissions in areas of Britain with population densities of

1 person or less. This approach to correct the VIIRS DNB for albedo and skyglow has previously been validated [76,77]. The generation of the light emission variable is summarized in Figure 1.

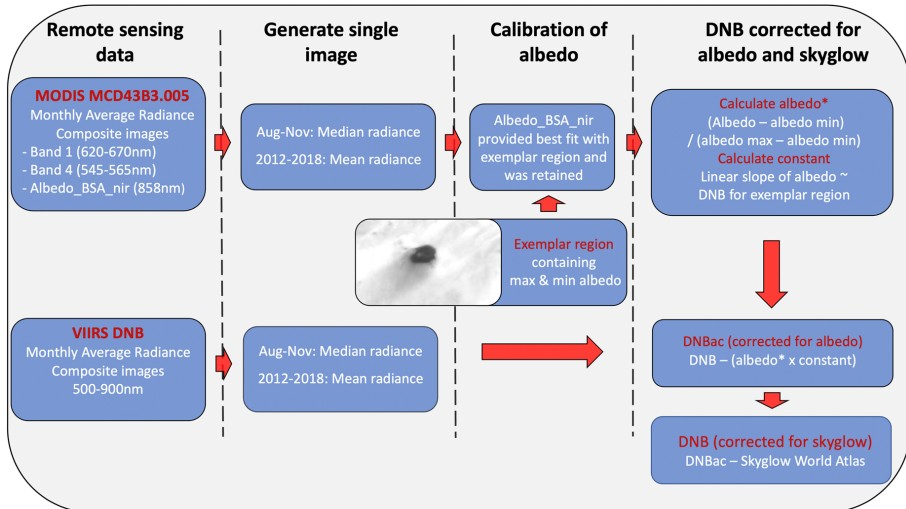

**Figure 1.** Flow diagram illustrating how the day/night band (DNB) layer corrected for albedo and skyglow was generated.

### 2.2.2. Skyglow

Estimates of variation in skyglow were obtained from Falchi et al. [59]. This surface was produced by modelling measured upward radiance from artificial sources, as measured by the DNB sensor and ground measurements. The skyglow atlas was generated for data collected in May, June, September, October, November and December in 2014 [59]. There was thus a mismatch with months selected for the VIIRS data, which were based on [72]. However, these months have been shown to have a good fit with sky brightness [76]. The data are presented for the global surface at a spatial resolution of 30 arc-seconds (c.1 km), and are the values for the approximate total brightness of the zenith radiance (mcd/m$^2$).

### 2.2.3. Human Population, Landcover, National Parks, SSSI and the Road Network

Data were obtained for five variables that were potentially correlated with the levels of light emissions and skyglow: human density, type of landcover, and whether the pixel was located within a National Park, a Site of Special Scientific Interest (SSSI), or it contained a road.

(i) Human density (continuous numerical): estimates of adjusted human density were obtained from the Gridded Population of the World (version 4; [78]). This product models the distribution of the human population on a continuous global raster at a resolution of 0.0083 degrees.

(ii) Landcover (categorical): we determined landcover using the GB 1 km dominant cover target aggregate classes raster product [79]. This is derived from a vector layer, reporting the aggregated habitat class with the highest percentage cover for each 1 km × 1 km pixel. The layer contains ten classes that are the aggregate of 21 target classes produced by the Joint Nature Conservation Committee Broad Habitats. We further aggregated classes to six categories based on broad similarities in structure and type (Table S1): broadleaf woodland and coniferous woodland were aggregated to 'woodland'; semi-natural grassland to 'grassland'; saltwater and freshwater to 'water'. Retained original classes were: mountain, heath and bog, coastal (rock/sediment), and built-up areas and gardens (renamed here as urban; see Table S1). To match the resolution of the other layers, following projection to the British National Grid the layer was resampled (nearest neighbour method) using the 'resample' function.

(iii) National Park (binomial): National Parks in Britain do not follow the international definition of such areas but are nationally significant landscapes, in which housing and commercial activities

are permitted, but subject to planning constraints. To determine whether pixels were located within a National Park or not, we downloaded a vector shapefile containing polygons delineating the National Park boundaries for mainland Britain [80]. We then applied the 'rasterize' function to generate a binary raster, where pixels were assigned a value of 1 if a National Park polygon covered the center of the pixel, and 0 if they did not.

(iv) Sites of Special Scientific Interest (SSSI) (binomial): SSSI are the primary legally designated protected areas in Britain. We obtained vector shapefiles of the boundaries of the SSSI in England [81], Wales [82] and Scotland [83]. These layers were joined to create a single layer of SSSI sites for Britain. We excluded solely geological sites, which may not be exposed above ground. The resulting layer of biological only sites and sites containing a mixture of biological and geological features (22,052 polygons from 8189 sites) was then rasterized, assigning a value of 1 to pixels if a SSSI polygon covered the center of a pixel, and 0 where they did not.

(v) Road network (categorical): We downloaded a vector shapefile of the road network of Britain from Digimap [84]. The shapefile metadata contains information on road type, which was used to generate a new polygon layer containing only major roads (A-roads, defined as major roads within, and linking urban centers, whilst having an average daily traffic flow of >20,000 vehicles; [85]) and motorways. A-roads and motorways account for 4% of England's road network and 43% of all traffic [85]. The resulting layer was rasterized to three categories representing pixels containing no road or a minor road, a major road or a motorway.

We designated each pixel as rural or urban, where a pixel was considered rural if it had a population density of ≤340 people per km$^2$, and urban >340 people per km$^2$. The summed population of rural and urban pixels was 17% and 83% of the total population of mainland Britain [65], respectively. For rural and then urban pixels, we calculated the light emissions per pixel/human population per pixel and the skyglow per pixel/human population per pixel.

## 2.3. Statistical Analysis

We stacked the seven resulting rasters, before converting them to a spatial points data frame containing the longitude and latitude of the centroids of each pixel, and a column for each variable (n = 450,545). We log-transformed the response variables (i.e., light emissions and skyglow), so that they were approximately normally distributed, before building a spatially lagged dependent variable model (also known as an SLX model), using the lmSLX command in the 'spdep' package. Spatially lagged models such as SLX allow for spatial local spillover effects, which are defined as the marginal impact of a change to one explanatory variable in a particular cross-sectional unit on the dependent variable values in another unit [86]. Therefore, SLX models provide coefficients for the effect of independent variables on the response ('direct' effect), and coefficients for the mean effect of the values of the independent variables of the nearest neighbors of each response ('indirect' effects). For example, in this case, along with modelling the direct influence of human density on the response, the SLX model also accounts for an indirect effect of the mean human density of neighboring pixels. The model took the following form:

$$\gamma = \chi\beta + \omega\chi\theta + \varepsilon \tag{3}$$

where $\gamma$ was either light emissions or skyglow. $\chi\beta$ represents the independent variables and coefficients, namely: human density, landcover, National Parks, SSSI and the road network. The spatial weights matrix is $\omega\chi\theta$, which is the average value of independent variables of the nearest neighbors and $\varepsilon$ is the spatially independent error.

To ensure computation times were tractable and to reduce spatial autocorrelation, SLX models were run on randomly selected subsets of 10% of the full dataset (45,054 pixels). We first used the *dnearneigh* function in the 'spdep' package to create a spatial weights matrix $\omega$ for the nearest neighbors within 10 km of the centroid of each pixel, thus allowing all pixels within this distance to influence the values in a pixel. To control for spatial autocorrelation in the model, we built correlograms of the SLX model residuals for the light emissions model and skyglow model using five different subsets of data.

We plotted correlograms for Moran's I at different multiples of the nearest neighbor distance, in this case equivalent to 10 km, 20 km, 30 km, etc, up to 150 km. For each response, we then selected the mean nearest neighbor distance across the five subsets, at which Moran's I was <0.03 (light emissions = 30 km and skyglow = 130 km). We then created new spatial weights matrices $\omega$ for the nearest neighbors within 30 km for light emissions and 130 km for skyglow, from the centroid of each pixel. For each response, we simulated the SLX model on 200 different randomly selected subsets of 10% of the full dataset. Model residuals for the light emission and skyglow models were approximately normally distributed (Figure S2). Finally, we used the impacts command in the spdep package to estimate the overall effect of both the direct and indirect effects (i.e., total coefficients). We present the mean coefficients, the standard error of the mean coefficients and the percentage of repetitions in which the *p* value <0.05.

## 3. Results

Across mainland Britain, median skyglow was greater than median light emissions, being driven by high levels of skyglow in England (Table 1a; Figure 2). Median light emissions per person were three times greater, and median skyglow per person 11 times greater, in the rural population, compared to the urban population (Table 1b). There were high levels of correlation between light emissions, skyglow and human density (Table 1c).

**Table 1.** Median light emissions and skyglow, (**a**) on mainland Britain and its constituent countries, and (**b**) per person in rural and urban areas. (**c**) The Spearman's rank correlations between light emissions, skyglow and human density. The interquartile range is given in parentheses (lower quartile, upper quartile).

| | Light Emissions (mcd/m$^2$) | Skyglow (mcd/m$^2$) |
|---|---|---|
| (a) Median | | |
| Mainland UK | 0.13 (0.08, 0.42) | 0.15 (0.04, 0.39) |
| England | 0.21 (0.11, 0.86) | 0.27 (0.14, 0.59) |
| Wales | 0.14 (0.11, 0.29) | 0.07 (0.03, 0.19) |
| Scotland | 0.08 (0.06, 0.10) | 0.02 (0.01, 0.09) |
| (b) Median per person | | |
| Rural per person | 0.0178 (0.008, 0.057) | 0.013 (0.006, 0.028) |
| Urban per person | 0.0057 (0.004, 0.009) | 0.0012 (0.001, 0.002) |
| (c) Spearman's rank correlation | | |
| Human density | 0.71 | 0.73 |
| Light emissions | - | 0.83 |

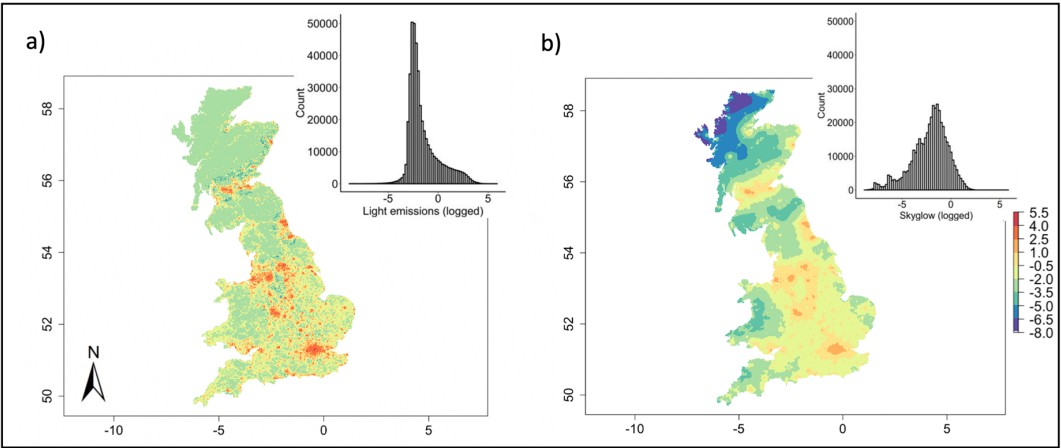

**Figure 2.** Spatial variation in (**a**) light emissions (logged) and (**b**) skyglow (logged) on mainland Britain. Histograms show the number of pixels of different intensities in bands of 0.2. Projected on the British National Grid, with a resolution of 544 m by 927 m.

*3.1. Light Emissions*

Median 95% confidence intervals for SLX light emission model coefficients for direct, indirect and total effects were within 0.6%, 3.2% and 12.8% of the mean (see Figure S1a), respectively, demonstrating a high degree of consistency for the direct and indirect effects. Across all models, the direct and indirect predictors in the SLX model explained a high proportion of the variance, with an average pseudo R-squared ($pR^2$) of 0.6.

When the direct effects were considered, there was a positive relationship between light emissions and human density, showing that when only the focal pixel is considered, light emissions increase with human density (Figure 3ai). However, when considering indirect effects (namely, the mean value of human density of neighboring pixels), there was a negative relationship, demonstrating that the light emissions per person decreases with increasing local human density (Table 2a). Total effects found a marginal negative relationship with human density (Table 2a). Although the direct effects for landcover demonstrated that light emissions were the least from mountain, heath and bog, indirect and total effects were least in woodland habitat, and all effects were greatest from urban areas (Table 2a; Figure 3b).

**Table 2.** Associations between (**a**) light emissions or (**b**) skyglow and five potential correlated variables on mainland Britain. We show the direct (effect of independent variables on the response), indirect (mean effect of independent variables of the nearest neighbor areas on the response) and total (direct and indirect combined) unstandardized coefficients. Spatial weights distance for light emissions was 30 km and for skyglow was 130 km. Factor variables are against a base factor level, i.e., for landcover the base factor level is mountain, heath, bog, thus, a positive coefficient suggests that other habitats receive more artificial light at night; the base factor levels for other variables are: National Parks, pixel is outside of a National Park; SSSI, pixel is outside of a SSSI; roads, no road or minor roads. We simulated the model 200 times for each response variable, and we present the mean coefficient and standard error of the mean. Statistical significance is shown (†††　>90% of models had a P value <0.05; †† > 75–90% of models; †60–75% of models; variables which had a p-value of >0.05 in <60% of models were considered to have marginal effects). We show the mean pseudo r squared across all models.

| | Direct Coefficients | Indirect Coefficients | Total Coefficients |
|---|---|---|---|
| (a) *Light emissions* | | | $R^2 = 0.63$ |
| Human density | $7.0 \times 10^{-4}$ ($\pm 3.0 \times 10^{-6}$)††† | $-1.3 \times 10^{-3}$ ($\pm 2.3 \times 10^{-5}$)††† | $-6.4 \times 10^{-4}$ ($\pm 2.2 \times 10^{-5}$)† |
| Landcover | | | |
| Woodland | 0.1 ($\pm 0.002$)††† | $-0.2$ ($\pm 0.001$)†† | $-0.1$ ($\pm 0.01$) |
| Grassland | 0.2 ($\pm 0.001$)††† | 1.0 ($\pm 0.007$)††† | 1.2 ($\pm 0.007$)††† |
| Water | 0.5 ($\pm 0.004$)††† | $-0.2$ ($\pm 0.04$) | 0.7 ($\pm 0.006$) |
| Coastal | 0.5 ($\pm 0.004$)††† | 3.2 ($\pm 0.05$)††† | 3.7 ($\pm 0.05$)††† |
| Urban | 1.9 ($\pm 0.004$)††† | 7.4 ($\pm 0.09$)††† | 9.4 ($\pm 0.08$)††† |
| National Parks | | | |
| Inside National Parks | $-0.4$ ($\pm 0.002$)††† | 0.2 ($\pm 0.004$)††† | $-0.2$ ($\pm 0.003$)††† |
| SSSI | | | |
| SSSI | $-0.2$ ($\pm 0.004$)††† | 0.8 ($\pm 0.01$)††† | 1.5 ($\pm 0.06$)††† |
| Roads | | | |
| Major road | 0.7 ($\pm 0.001$)††† | 0.8 ($\pm 0.06$)†† | 1.2 ($\pm 0.06$)††† |
| Motorway | 1.3 ($\pm 0.005$)††† | 4.1 ($\pm 0.1$)††† | 5.4 ($\pm 0.1$)††† |
| (b) *Skyglow* | | | $R^2 = 0.75$ |
| Human density | $5.0 \times 10^{-4}$ ($\pm 1.5 \times 10^{-6}$)††† | $-2.2 \times 10^{-3}$ ($\pm 2.4 \times 10^{-4}$)†† | $-1.7 \times 10^{-3}$ ($\pm 2.3 \times 10^{-4}$)†† |
| Landcover | | | |
| Woodland | 0.4 ($\pm 0.002$)††† | 3.3 ($\pm 0.02$)†† | 3.7 ($\pm 0.2$)†† |
| Grassland | 0.5 ($\pm 0.002$)††† | 1.2 ($\pm 0.07$)††† | 1.7 ($\pm 0.07$)††† |
| Water | 0.6 ($\pm 0.008$)††† | $-57.1$ ($\pm 1.5$)††† | $-56.5$ ($\pm 1.5$)††† |
| Coastal | 0.9 ($\pm 0.003$)††† | $-50.5$ ($\pm 3.2$)††† | $-49.5$ ($\pm 3.2$)††† |
| Urban | 1.4 ($\pm 0.003$)††† | 30.3 ($\pm 0.9$)††† | 31.7 ($\pm 0.9$)††† |
| National Parks | | | |
| Inside National Parks | $-0.5$ ($\pm 0.002$)††† | 4.6 ($\pm 0.06$)††† | 4.1 ($\pm 0.06$)††† |
| SSSI | | | |
| SSSI | $-0.1$ ($\pm 0.002$)††† | $-10.8$ ($\pm 0.2$)††† | $-10.9$ ($\pm 0.2$)††† |
| Roads | | | |
| Major road | 0.3 ($\pm 0.001$)††† | $-8.4$ ($\pm 0.7$)††† | $-8.1$ ($\pm 0.7$)††† |
| Motorway | 1.0 ($\pm 0.004$)††† | $-50.7$ ($\pm 1.6$)††† | $-49.7$ ($\pm 1.6$)††† |

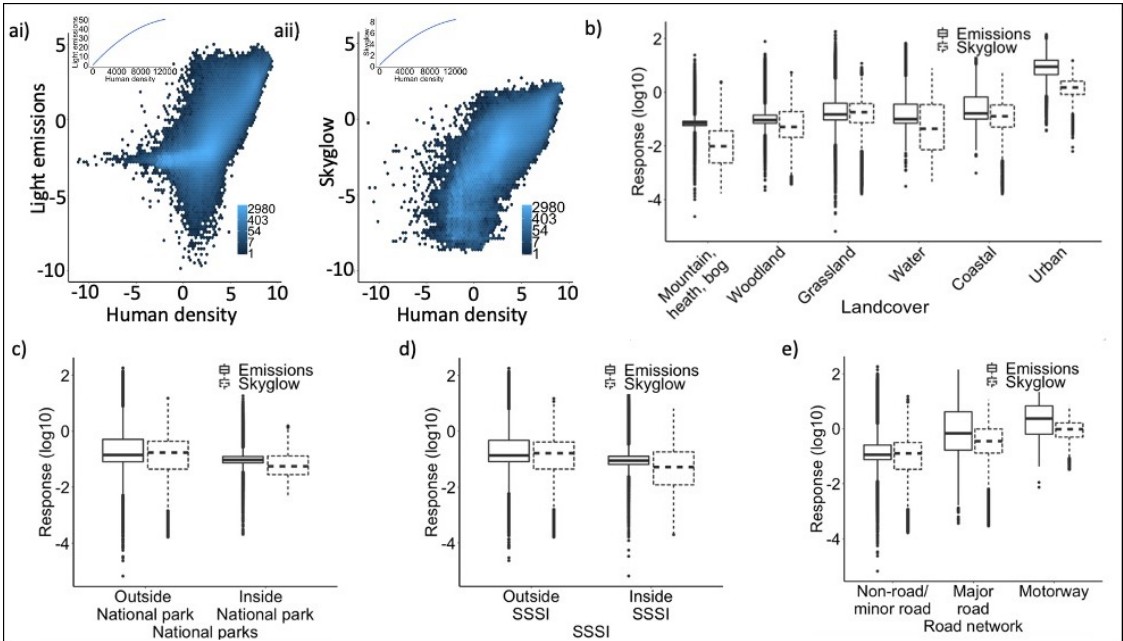

**Figure 3.** The relationship between (**i**) light emissions and (**ii**) skyglow with (**a**) human density (logged), (**b**) landcover, (**c**) National Parks, (**d**) SSSI, and (**e**) roads. (**a**) gives the logged values for light emissions and skyglow, while (**b–e**) show both values at log10. The legend for (**a**) shows the log of the number of pixels, and the plot in the top left of the panels illustrates the asymptotic relationship. Boxplots (**b–e**) show the median, the lower and upper quartiles and the minimum and maximum values, and the legend denotes whether the data illustrates the relationship with light emissions (solid line) or skyglow (dashed line).

When total and direct effects were considered there were fewer light emissions from within National Parks than from outside, whilst there was a positive relationship with indirect effects (Table 2a; Figure 3c). Direct effects showed that there were fewer light emissions from inside SSSI sites than from outside, whilst there was a positive relationship with indirect and total effects (Table 2a; Figure 3d). There was a gradient of least light emissions from no roads or minor roads, to most light emissions associated with areas containing motorways, with the total effects from pixels containing motorways being five times greater than from those containing major roads (Table 2a; Figure 3e).

### 3.2. Skyglow

All 95% confidence intervals for SLX light emission model coefficients for direct effects, indirect effects and total effects were within 0.4%, 16.5% and 21.4%, of the mean (see Figure S1b), respectively, demonstrating a high degree of consistency for the direct and indirect effects. Across models the direct and indirect predictors explained a high proportion of the variance, with an average pR$^2$ of 0.75.

When considering the direct effects, there was a positive relationship between skyglow and human density (Figure 3b), whilst, when considering the indirect and total effects (direct and indirect effects combined), the relationship between skyglow and human density was negative (Table 2b). Direct effects showed that, for landcover, levels of skyglow were least in areas classed as mountain, heath and bog, and greatest in urban areas (Figure 3b), whilst indirect and total effects were least in coastal habitats and greatest in urban areas (Table 2b).

There was a negative relationship between levels of skyglow and whether a location was in a National Park or not (Figure 3c), however, when considering the indirect effects and total effects, there was a positive relationship between skyglow and National Parks (Table 2b). We found that the direct effects, indirect effects and total effects showed that the levels of skyglow were less within SSSI than outside (Table 2b; Figure 3d). There were lower levels of skyglow associated with areas containing

no roads or minor roads, and the greatest level of skyglow associated with areas containing motorways (Table 2b; Figure 3e). Direct effects showed that skyglow was twice as great in areas with motorways than those with major roads (Table 2b). However, when considering the indirect and total effects, skyglow was greatest from areas with no roads or minor roads, and least from those with major roads and motorways (Table 2b).

## 4. Discussion

At a spatial resolution of c.500,000 m$^2$, the variation in both artificial nighttime light emissions and skyglow is marked. Although levels of the two are spatially correlated, inevitably the distribution of light emissions is more localized, whilst some level of skyglow was detected over the entirety of Britain. This latter observation is consistent with recent evidence that skyglow can extend hundreds of kilometers from urban sources (e.g., [87]). It is also reflected in the longer distances over which a marked spatial autocorrelation was detected in skyglow compared to light emissions.

The maximum measured light emissions were more than 40 times greater than maximum modelled skyglow. To calculate measured light emissions for each pixel, we subtracted the modelled values of skyglow [59] from the light emissions corrected for albedo. In green spaces located on the outskirts of large urban centers (known in the UK as the green belt), there were high levels of skyglow but few light emissions, which led us to calculate negative values for light emissions (which were subsequently corrected to 0.08 mcd/m$^2$). This suggests that the values of skyglow modelled by [59] overestimate the degree of skyglow around urban centers, possibly because a constant value of atmospheric aerosols was used, although these might vary markedly within and beyond urban areas. This overestimation of skyglow is likely to be applicable across the entire skyglow atlas. Median levels of light emissions were greatest in England, with levels in Wales being two-thirds and levels in Scotland a third that of England. Only 7.5% of people in Britain are able to see the Milky Way where they live due to artificial light at night, because levels of both light emissions (8.3% of population) and skyglow (20.5% of population) are below 0.6 mcd/m$^2$.

Much of the spatial variation in levels of light emissions and skyglow was accounted for in models by the independent variables analyzed (human density, landcover, National Parks, SSSI and roads). Landcover was the strongest predictor of both dimensions of artificial light at night, followed by human density for light emissions and the road network for skyglow (Table S2). Across Britain, we found an asymptotic relationship between the levels of light emissions or skyglow and human density (Figure 3a), with the amount of light emissions per person decreasing with human density. This confirms work by previous studies at the national and regional scale [88,89], and is supported here by the negative relationship with the indirect effects in the spatial model, which demonstrated that the more people that live in an area, the less artificial light is produced per person. Indeed, we found that the median light emissions per person were three times greater in rural than urban areas, and skyglow 11 times greater. Overall, we found that the indirect effects were stronger than the direct effects (indicated by a negative total effects), demonstrating that, although there was a positive relationship between human density and artificial light at night, there was a stronger effect of people living in close proximity in urban areas producing less light per person. Thus, resources are used more efficiently than in rural areas (for example, there can be significant variation in the number of people living and working around roads with similar densities of streetlights; [90]). This can be clearly seen in England, where 84% of people live in urban areas ([91]). Whereas, for example, in Scotland, which has a more rural population, if modelled in isolation, there were no significant indirect effects between light emissions and human density (Direct $1.5 \times 10^{-3} \pm 4.8 \times 10\text{-}5$; Indirect $-6.1 \times 10^{-4} \pm 7.1 \times 10^{-4}$), while the median levels of both light emissions and skyglow per person were twice those of England or Wales.

As one might expect, the direct effects showed that artificial light at night was least in mountains, heath and bog and greatest in urban areas. The median levels of light emissions were higher than median skyglow for all landcover types, except for grassland and croplands. Central and south England

have high degrees of skyglow, while also containing large areas of arable, improved grassland and semi-natural grassland (here termed grassland; Table S1), habitats characterized by short vegetation structure and large fields while also often being close to urban infrastructure. Indirect and total effects showed that the contribution of neighboring pixels to direct emissions in the focal pixel was least in woodland areas, which is likely due to reduced infrastructure in surrounding woodland shown by reduced direct effects, and because trees physically block the spread of light, thereby diminishing the area it can penetrate into. Indirect effects found that skyglow was least at coastal sites and inland and marine water, and these sites tend to be relatively far from urban areas, and are often surrounded by mountainous topography that is not conducive for the spread of artificial light at night, so these sites have a minimal influence on the focal pixel. The behavior of artificial light at night in different landcover types provides more detailed information on previously reported global trends in different ecoregions [48].

There were lower levels of artificial light at night inside of National Parks than outside (shown by negative direct effects). Although the overall levels of light emissions were minimal, we found that there were a number of strong sources of artificial light inside these areas (Figure 3c). In the UK, National Parks contain villages, towns and industry, although these are subject to planning restrictions, and this result suggests that National Park planning acts to constrain light pollution. Conversely, there was a positive relationship for indirect effects, demonstrating that mean higher levels of light emissions and skyglow in pixels surrounding National Parks are associated with increased skyglow inside of National Parks. This relationship is driven by National Parks in England, which can be surrounded by high levels of urbanization. Indeed, the median skyglow was higher than the median light emissions in five of the ten National Parks in England; in the Peak District National Park, for example, an upland area in the North of England surrounded by major cities, the median skyglow was an order of magnitude greater than median light emissions (see Table S3). The median skyglow within the National Parks suggests that, on average, it is possible to see the Milky Way in winter (when it is faintest) in all National Parks (skyglow levels are below 0.6 mcd/m$^2$, [59]), with the best views in the Cairngorms National Park in Scotland (0.01 mcd/m$^2$), although the Milky Way is likely to be faint in the Peak District (0.44 mcd/m$^2$) National Park. A cause for concern is that the newly designated South Down National Park Dark Skies Reserve had the third highest median levels of skyglow of the 15 National Parks on mainland Britain. The UK is indicative of an increasing global issue of the pervasiveness of artificial light at night and, in particular, skyglow into key biodiversity areas [92].

Direct effects show that both forms of artificial light at night were lower within SSSI than outside. Further, indirect effects for light emissions show that if the mean light emissions are higher in pixels neighboring a SSSI pixel, then there will be lower levels of artificial light in the SSSI pixel, demonstrating that light emissions in and around SSSI tend to be localized. Conversely, if there is less skyglow in pixels neighboring a SSSI pixel, then there will less skyglow inside the SSSI pixel, thereby reflecting the greater dispersion and reduced localization of skyglow compared to direct emissions. A cause for concern is that there still appear to be powerful sources of artificial light at night within or immediately adjacent to (i.e., in the same pixel) some SSSIs (Figure 3d). These are likely to have significant biological impacts on the ecology of these protected spaces, and are prime locations for targeted management to reduce such impacts.

Modelled direct effects demonstrate that the presence of major roads and motorways are significant contributors to artificial light at night, with motorways emitting two or three times more artificial light at night than major roads. Indirect effects of light emissions show that the higher light emissions in neighboring pixels are associated with higher light emissions in the focal road pixel, demonstrating how, over short distances (30 km neighborhood object in this case), the associated road and urban infrastructure magnify the overall levels of light emissions. Conversely, for skyglow, there was a negative relationship for indirect effects for major roads and motorways, demonstrating that if skyglow is lower in areas surrounding a major road or motorway, then there will be more skyglow associated with the major road or motorway. This is a consequence of major roads and motorways passing through

rural areas that have reduced levels of skyglow relative to the surrounding urban areas. Further, to control for spatial autocorrelation, the neighborhood object for skyglow was 130 km, and therefore the mean skyglow of the neighboring pixels will be lowered by the inclusion of larger rural areas, making bright roads brighter. Importantly, artificial lights from vehicles tend to be powerful, usually projected in the horizontal plane and are temporally inconsistent, and therefore are unreliably detected by satellite imagery (see [93]). The contribution of the road network is thus likely to be greater than detected here.

The VIIRS DNB is the most popular and sensitive remote sensing data available for the study of artificial light at night, however, there are several limitations to its use. The first is that it does not detect light in the blue and violet spectrum, and no public data are currently available that could be used to resolve this issue. Globally, there has been an increasing trend of switching away from traditional sodium lamps to white LEDs, particularly in urban areas, due to their perceived increased energy efficiency [94]. As a consequence, the actual amount of artificial light at night is likely to be greater than estimated here, and this may reduce the difference between the light emissions produced per person in rural and those produced per person in urban areas. A cause for concern is that there is increasing awareness amongst ecologists of the broader negative environmental impacts of white LEDs compared to sodium lamps [3]. A second limitation is that VIIRS captures images at c.01:30 local time, a period when many ornamental and commercial lights, along with vehicle traffic, are close to their minimum intensity. Therefore, the use of VIIRS data can be thought to provide a baseline or lower limit of artificial light at night. Third, while there has been much interest in the development of standardized measurement systems from the ground to verify remote sensing nighttime lighting data [95–99], the collection of these data in the UK was outside of the scope of this study, and would constitute a major program of work given the size and heterogeneity of the region, and the challenges of addressing the spectral issues highlighted above. Indeed, to date, projects to ground truth remotely sensed artificial lighting data have not resolved this last issue [96,97,99]. In future, it would be interesting to compare how spatial relationships vary through time, but perhaps in a more dynamic region. It is also important to note that cloud cover is known to amplify the intensity of skyglow, as seen from the ground, and to influence its spatial distribution [100], however, no data are currently available to incorporate cloud cover into measures of skyglow, as calculated by the World Atlas of Artificial Night Sky Brightness. To inform policy and planning, future research needs to focus on determining at the regional level the more detailed spatial distribution of all artificial light at night, including more temporary, mobile forms, such as vehicles, and how these vary both with changing climatic conditions, such as cloud cover and across the course of the nighttime.

## 5. Conclusions

Britain is one of a growing number of regions of the world in which virtually no areas are left that experience natural light regimes. The analyses reported here emphasize the pervasiveness of artificial nighttime light, especially when both light emissions and skyglow are considered. The spatial patterns of this light are richly textured, dependent on combinations of local and wider sources. The higher per capita emissions in rural than urban areas are particularly striking. Urban areas have been the focus of significant resources aimed at reducing the levels of artificial light at night, and we show that in terms of artificial light at night, per capita they are more energy efficient and produce less pollution [90]. From a resource management perspective, our results suggest that this approach should be matched with exploration of ways of reducing artificial light at night in rural areas. Differences in artificial light levels associated with different land uses (e.g., less in mountains and woodlands, much more in urban areas) are expected, but the wide variance in levels associated with non-urban land uses is notable, and highlights how far artificial lighting has come from being an exclusively urban issue. Likewise, comparisons within and outside of protected areas and protected landscapes reveal that these are regularly failing markedly to buffer nature from any pressures that artificial lighting poses [101]. From a global perspective, understanding the different land uses and structures that are

associated with artificial light at night will inform governments, international, national and local stakeholders where best to target resources to reduce their negative human health and ecological impacts and focus artificial light at night to where, when and in the form that is required.

**Supplementary Materials:** The following are available online at http://www.mdpi.com/2072-4292/12/10/1591/s1, Figure S1: Variation in the coefficient of independent variables for (**a**) light emissions and (**b**) skyglow, Figure S2: Histograms and qqplots of model residuals for (**a**) light emissions, and (**b**) skyglow, Table S1: Relationship between the aggregated classes used here, the landcover map aggregated classes and broad habitat, Table S2: Relative importance of each variable in explaining levels of (**a**) light emissions, and (**b**) skyglow, Table S3: Levels of (**a**) light emissions and (**b**) skyglow inside of National Parks in Britain.

**Author Contributions:** Conceptualization, D.T.C.C., A.S.d.M. and K.J.G.; Data curation, D.T.C.C., A.S.d.M. and S.A.D.; Formal analysis, D.T.C.C. and J.B.; Funding acquisition, K.J.G.; Methodology, J.B.; Software, A.S.d.M. and S.A.D.; Supervision, K.J.G.; Visualization, D.T.C.C. and A.S.d.M.; Writing—original draft, D.T.C.C. and K.J.G.; Writing—review & editing, D.T.C.C., A.S.d.M., J.B. and K.J.G. All authors have read and agreed to the published version of the manuscript.

**Funding:** This work was supported by the EMISSI@N project funded by Natural Environment Research Council grant NE/P01156X/1.

**Acknowledgments:** We thank Richard Sherley for statistical discussions.

**Conflicts of Interest:** The authors declare no conflict of interest. The funders had no role in the design of the study; in the collection, analyses, or interpretation of data; in the writing of the manuscript, or in the decision to publish the results.

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
