# Peer review of "National Scale Spatial Variation in Artificial Light at Night"

_remotesensing, doi:10.3390/rs12101591_

Round 1

Reviewer 1 Report

Overall comments:

The subject of the article is highly relevant as artificial light at night (ALAN) is increasing continuously and changing nightscapes with unforeseen consequences for human and nature. The manuscript nicely introduces the topic and offers a very detailed and transparent description of the methods. However, the data collection of remote sensing via satellites should be described in more detailed naming all insufficiencies. Besides the lack in sensitivity of the blue light the satellite data is recorded at a single time point every day, light sources that are used outside of the time window of the satellite crossing, will not be detected. The conversion to modern, efficient lighting increases emissions in the blue part of the spectrum under 500nm wavelengths. Thus the results might be biased by a higher rate of street light modernisation in urban and peri-urban areas, compared to rural sides.

The authors should explain why they did not apply further measurements from the ground to verify the remote sensing data. The development of standardized measurement systems of ALAN is rather an important subject and should not be neglected. I highly recommend adding a discussion of the existing ground-based measurement of ALAN developed to support remote sensing data. The discussion of ground-based measurement is particularly important for future research because a standardized measurement of sky brightness and skyglow is urgently required to protect nightscapes, species and ecosystem. Following references are recommended to be considered for this discussion:

  • Barentine, J. C. (2019). Methods for assessment and monitoring of light pollution around ecologically sensitive sites. Journal of Imaging, 5(5), 54.
  • Jechow, A., Ribas, S. J., Domingo, R. C., Hölker, F., Kolláth, Z., & Kyba, C. C. (2018). Tracking the dynamics of skyglow with differential photometry using a digital camera with fisheye lens. Journal of Quantitative Spectroscopy and Radiative Transfer, 209, 212-223.
  • Hänel, A., Posch, T., Ribas, S. J., Aubé, M., Duriscoe, D., Jechow, A., ... & Spoelstra, H. (2018). Measuring night sky brightness: methods and challenges. Journal of Quantitative Spectroscopy and Radiative Transfer, 205, 278-290.
  • Wallner, S. (2019). Usage of Vertical Fisheye-Images to Quantify Urban Light Pollution on Small Scales and the Impact of LED Conversion. Journal of Imaging, 5(11), 86.
  • Kocifaj, M., Wallner, S., & Solano-Lamphar, H. A. (2019). An asymptotic formula for skyglow modelling over a large territory. Monthly Notices of the Royal Astronomical Society, 485(2), 2214-2224.

Furthermore, it should be stated that the data are a fixed assumption of the amount and distribution of ALAN before 2016. It would be nice to describe the methods of Falchi et al. (2016) and mention the contributions of citizen scientist data in connection with remote sensing tools. The increase of ALAN is estimated by about 2-6% per year and changes might impact on the results of the present study, because ALAN increase is not equally distributed. For example, the changes of ALAN increase in Germany from 2012-2016 present rather great differences between the federal states. The results of the German study might correlate or contradict with the results of the present study, as the German federal states present high differences in population densities, distribution of urban and rural landscapes, respectively. It is recommended to discuss the following reference or similar studies in this context: Kyba, C., Kuester, T., & Kuechly, H. (2017). Changes in outdoor lighting in Germany from 2012-2016. International Journal of Sustainable Lighting, 19(2), 112-123.

Minor comments:

Lines 78-82 either part of results or to be phrased as hypothesis.

Lines 422-433 please refer to: Kyba, C. C., Ruhtz, T., Fischer, J., & Hölker, F. (2011). Cloud coverage acts as an amplifier for ecological light pollution in urban ecosystems. PloS one, 6(3).

Lines 442-444 recommended to cite: Schroer, S., Huggins, B. J., Azam, C., & Hölker, F. (2020). Working with Inadequate Tools: Legislative Shortcomings in Protection against Ecological Effects of Artificial Light at Night. Sustainability, 12(6), 2551.

Author Response

Reviewer 1

Overall comments:

REVIEWER: The subject of the article is highly relevant as artificial light at night (ALAN) is increasing continuously and changing nightscapes with unforeseen consequences for human and nature. The manuscript nicely introduces the topic and offers a very detailed and transparent description of the methods.

RESPONSE: We thank the reviewer for their encouraging comments.

REVIEWER: However, the data collection of remote sensing via satellites should be described in more detailed naming all insufficiencies. Besides the lack in sensitivity of the blue light the satellite data is recorded at a single time point every day, light sources that are used outside of the time window of the satellite crossing, will not be detected. The conversion to modern, efficient lighting increases emissions in the blue part of the spectrum under 500nm wavelengths. Thus, the results might be biased by a higher rate of street light modernisation in urban and peri-urban areas, compared to rural sides.

RESPONSE: We have now expanded the limitations section in the discussion to also include these two points.

Text added L515-L527: The VIIRS DNB is the most popular and sensitive remote sensing data available for the study of artificial light at night, however there are several limitations to its use. The first is that it does not detect light in the blue and violet spectrum, and no public data are currently available that could be used to resolve this issue. Globally, there has been an increasing trend of switching away from traditional sodium lamps to white LEDs particularly in urban areas, due to their perceived increased energy efficiency (Traverso et al., 2017). As a consequence, the actual amount of artificial light at night is likely to be greater than estimated here, and this may reduce the difference between the light emissions produced per person in rural and those produced per person in urban areas. Of concern is that there is increasing awareness amongst ecologists of the broader negative environmental impacts of white LEDs compared to sodium lamps (Gaston et al. 2014). A second limitation is that VIIRS captures images at c.01:30 local time, a period when many ornamental and commercial lights, along with vehicle traffic are close to their minimum intensity. Therefore, the use of VIIRS data can be thought to provide a baseline or lower limit of artificial light at night.

Traverso, M.; Donatello, S.; Moons, H.; Rodriguez Quintero, R.; Gama Caldas, M.; Wolf, O.; Van Tichelen, P.; Van Hoof, V.; Geerken, T.  Revision of the EU green public procurement criteria for street lighting and traffic signals. Publications Office of the European Union, 2017, Luxembourg. Doi: 10.2760/479108

Gaston, K.J.; Duffy, J.P.; Gaston, S.; Bennie, J.; Davies, T.W. Human alteration of natural light cycles: causes and ecological consequences. Oecologia. 2014, 176, 917–931. https://doi.org/10.1007/s00442-014-3088-2

REVIEWER: The authors should explain why they did not apply further measurements from the ground to verify the remote sensing data. The development of standardized measurement systems of ALAN is rather an important subject and should not be neglected. I highly recommend adding a discussion of the existing ground-based measurement of ALAN developed to support remote sensing data. The discussion of ground-based measurement is particularly important for future research because a standardized measurement of sky brightness and skyglow is urgently required to protect nightscapes, species and ecosystem. Following references are recommended to be considered for this discussion:

  • Barentine, J. C. (2019). Methods for assessment and monitoring of light pollution around ecologically sensitive sites. Journal of Imaging5(5), 54.
  • Jechow, A., Ribas, S. J., Domingo, R. C., Hölker, F., Kolláth, Z., & Kyba, C. C. (2018). Tracking the dynamics of skyglow with differential photometry using a digital camera with fisheye lens. Journal of Quantitative Spectroscopy and Radiative Transfer209, 212-223.
  • Hänel, A., Posch, T., Ribas, S. J., Aubé, M., Duriscoe, D., Jechow, A., ... & Spoelstra, H. (2018). Measuring night sky brightness: methods and challenges. Journal of Quantitative Spectroscopy and Radiative Transfer205, 278-290.
  • Wallner, S. (2019). Usage of Vertical Fisheye-Images to Quantify Urban Light Pollution on Small Scales and the Impact of LED Conversion. Journal of Imaging5(11), 86.
  • Kocifaj, M., Wallner, S., & Solano-Lamphar, H. A. (2019). An asymptotic formula for skyglow modelling over a large territory. Monthly Notices of the Royal Astronomical Society485(2), 2214-2224.

RESPONSE: We now discuss ground truthing of remote sensing data in the limitations section of the discussion.

TEXT ADDED L538-L545: Third, while there has been much interest in the development of standardized measurement systems from the ground to verify remote sensing data (Hänel et al., 2018; Jechow et al., 2018; Barentine, 2019; Kocifaj et al., 2019; Wallner, 2019), collection of these data in the UK was outside of the scope of this study, and would constitute a major program of work given the size and heterogeneity of the region and the challenges of addressing the spectral issues highlighted above. Indeed, to date projects to ground truthing remotely senses artificial lighting data have not addressed this last issue (Jechow et al., 2018; Barentine, 2019; Wallner, 2019).

Barentine, J. C. (2019). Methods for assessment and monitoring of light pollution around ecologically sensitive sites. Journal of Imaging5(5), 54.

Hänel, A., Posch, T., Ribas, S. J., Aubé, M., Duriscoe, D., Jechow, A., ... & Spoelstra, H. (2018). Measuring night sky brightness: methods and challenges. Journal of Quantitative Spectroscopy and Radiative Transfer205, 278-290.

Jechow, A., Ribas, S. J., Domingo, R. C., Hölker, F., Kolláth, Z., & Kyba, C. C. (2018). Tracking the dynamics of skyglow with differential photometry using a digital camera with fisheye lens. Journal of Quantitative Spectroscopy and Radiative Transfer209, 212-223.

Kocifaj, M., Wallner, S., & Solano-Lamphar, H. A. (2019). An asymptotic formula for skyglow modelling over a large territory. Monthly Notices of the Royal Astronomical Society485(2), 2214-2224.

Wallner, S. (2019). Usage of Vertical Fisheye-Images to Quantify Urban Light Pollution on Small Scales and the Impact of LED Conversion. Journal of Imaging5(11), 86.           

REVIEWER: Furthermore, it should be stated that the data are a fixed assumption of the amount and distribution of ALAN before 2016. It would be nice to describe the methods of Falchi et al. (2016) and mention the contributions of citizen scientist data in connection with remote sensing tools. The increase of ALAN is estimated by about 2-6% per year and changes might impact on the results of the present study, because ALAN increase is not equally distributed. For example, the changes of ALAN increase in Germany from 2012-2016 present rather great differences between the federal states. The results of the German study might correlate or contradict with the results of the present study, as the German federal states present high differences in population densities, distribution of urban and rural landscapes, respectively. It is recommended to discuss the following reference or similar studies in this context: Kyba, C., Kuester, T., & Kuechly, H. (2017). Changes in outdoor lighting in Germany from 2012-2016. International Journal of Sustainable Lighting19(2), 112-123.

RESPONSE: We now discuss the temporal increase in artificial light at night in the methods. To maintain the focus on the UK we cite Kyba et al. (2017) Artificial lit surface of Earth at night increasing in radiance and extent’ Sci adv 22 e1701528 Figure S26. We agree that it would be interesting to explore substructures of artificial light at night in the UK. However, it is outside of the scope of this study.

TEXT ADDED L160-L163: A single averaged measure of light emissions does not account for growth through time in the radiance of light emissions in the UK (Kyba et al., 2017). However, here were interested in spatial variation and although related to growth it is several orders of magnitude greater, and so temporal changes are not comparable.

Kyba, C.C.M.; Kuester, T.; De Miguel, A.S.; Baugh, K.; Jechow, A.; Hölker, F.; Bennie, J.; Elvidge, C.D.; Gaston, K.J.; Guanter, L. Artificially lit surface of Earth at night increasing in radiance and extent. Sci. Adv. 2017, 3(11), e1701528. doi: 10.1126/sciadv.1701528

Minor comments:

REVIEWER: Lines 78-82 either part of results or to be phrased as hypothesis.

RESPONSE: The paragraph has been phrased in line with the journal manuscript preparation guidelines ‘‘Finally, briefly mention the main aim of the work and highlight the main conclusions.” As such we have not altered the text.

REVIEWER: Lines 422-433 please refer to: Kyba, C. C., Ruhtz, T., Fischer, J., & Hölker, F. (2011). Cloud coverage acts as an amplifier for ecological light pollution in urban ecosystems. PloS one6(3).

RESPONSE: Citation included as suggested, many thanks.

REVIEWER: Lines 442-444 recommended to cite: Schroer, S., Huggins, B. J., Azam, C., & Hölker, F. (2020). Working with Inadequate Tools: Legislative Shortcomings in Protection against Ecological Effects of Artificial Light at Night. Sustainability12(6), 2551.

RESPONSE: Citation included as suggested, many thanks.

Reviewer 2 Report

At the start, the paper seems interesting, but after reading all the text, the paper turns out to be hard to read, does not introduce significant and exciting conclusions.

The authors averaged satellite images and used a modelled sky glow to build relations between seven layers (human and landcover) and direct emission or sky glow.

They use a logarithmic scale for emission and sky glow to make it "approximately normally distributed." There is no justification that the resultant distribution is similar to normal. In my opinion (based on figure 1), the distribution has nothing to do with a normal distribution.

The model of emission built as corrected DNB without sky glow is entirely untested. So, the reader can only trust that it can potentially be correct. 

The authors didn't comment on this issue in the text.

The whole process of calculation should be illustrated as a flowchart to make it understandable. The function used for calculations, lmSLX is deprecated and won't be present in the next release of the 'spdep' package (information from manual for Version 1.1-3, date 2019-09-18). The reader could not repeat calculations with the next release of the software.

Almost all tables are moved to supplementary materials. The authors put results (numbers) in the text, rather than to summarize it in a readable form. 

The results' analysis contains obvious observations. For example:

  • "Spearman's rank correlation showed that light emissions were correlated with skyglow and human population (r s = 0.83 and 0.71, respectively), while skyglow was also correlated with human population (r s =0.73)";
  • "Total effects, direct effects and indirect effects all showed greater light emissions from areas containing motorways than from those containing major roads, which in turn were greater than light emissions from areas containing no roads or minor roads (Table 1a; Figure 2e)";
  • "Direct effects for landcover demonstrated that light emissions were least from mountain, heath and bog and were greatest from urban areas (Table 1a; Figure 2b). When considering indirect and total effects light emissions were least in woodland habitat and greatest in urban areas (Table 1a)."

From a technical point of view, the text contains fragments of the template (conflict of interest), and the literature is unordered. It seems to be a draft rather than a final paper submitted to the journal. 

Conclusions:

The major part of the paper has to be rewritten. The whole processing path needs a good illustration and clear description. The results should be placed in simple tables with descriptions only in the text. 

The authors should add any justification and comment on the emission model. 

If authors would like to present the technical aspect of calculations (software, commands used, scripts, high-resolution maps), supplementary materials are the right place for it.

The authors should focus on new and interesting conclusions rather than on evident observations. 

The submission should contain the final paper that is ready to be read by editors, reviewers, and finally - readers.

Recommendation:

Major revision

Author Response

Reviewer 2

REVIEWER: At the start, the paper seems interesting, but after reading all the text, the paper turns out to be hard to read, does not introduce significant and exciting conclusions.

RESPONSE: We regret that the reviewer did not enjoy the manuscript. We note however that by contrast both reviewers 1 and 3 responded very positively, and did not share these concerns.

REVIEWER: The authors averaged satellite images and used a modelled sky glow to build relations between seven layers (human and landcover) and direct emission or sky glow. They use a logarithmic scale for emission and sky glow to make it "approximately normally distributed." There is no justification that the resultant distribution is similar to normal. In my opinion (based on figure 1), the distribution has nothing to do with a normal distribution.

RESPONSE: The distribution of both response variables was logged so that they were ‘approximately normally distributed’ for inclusion in the lmSLX models. More importantly from a modelling perspective the model residuals from both the light emission and skyglow models were normally distributed. We have now included a histogram and qqplot from each model in the supplementary information (Figure S2). To avoid confusion we have deleted the ‘approximately normally distributed’ from L221.

TEXT ADDED L274-275: ‘Model residuals for the light emission and skyglow models were approximately normally distributed (Figure S2).’

REVIEWER: The model of emission built as corrected DNB without sky glow is entirely untested. So, the reader can only trust that it can potentially be correct. The authors didn't comment on this issue in the text.

RESPONSE: The emission model of DNB corrected for albedo and skyglow has been validated in two studies by two different research groups (Sánchez de Miguel et al., 2019; Kocifaj & Bara 2020).

TEXT ADDED L151-L152: This approach to correct the VIIRS DNB for albedo and skyglow has previously been validated (Sánchez de Miguel et al., 2020; Kocifaj & Bará 2020).

Sánchez de Miguel, A.; Kyba, C.; Zamorano, J.; Gallego, J.; Gaston, K.J. The nature of the diffuse light near cities detected in nighttime satellite imagery. 2019arXiv preprint arXiv:1908.05482.

Kocifaj, M.; Wallner, S.; Solano-Lamphar, H.A. An asymptotic formula for skyglow modelling over a large territory. Mon. Not. R. Astron. Soc. 2019, 485(2), 2214-2224. https://doi.org/10.1093/mnras/stz520

REVIEWER: The whole process of calculation should be illustrated as a flowchart to make it understandable.

RESPONSE: We thank the reviewer for their suggestion. We now include a flow chart of how the VIIRS DNB layer corrected for albedo and skyglow was calculated as a new figure 1.

REVIEWER: The function used for calculations, lmSLX is deprecated and won't be present in the next release of the 'spdep' package (information from manual for Version 1.1-3, date 2019-09-18). The reader could not repeat calculations with the next release of the software.

RESPONSE: We thank the reviewer for bringing this to our attention. We have given the package version number in the text. The ‘spdep’ documentation ‘2019-09-18’ states that the function will still be available in the ‘spatialreg’ package (Bivand et al., 2013) and therefore readers will be able to repeat the analyses. It is also possible to install older versions of the ‘spdep’ package.

Bivand, R., G. Piras. Comparing implementations of estimation methods for spatial econometrics. J. Stat. Softw. 2015, 63, 1-36. doi: 10.18637/jss.v063.i18

REVIEWER: Almost all tables are moved to supplementary materials. The authors put results (numbers) in the text, rather than to summarize it in a readable form.

RESPONSE: We now include a new table 1 that summarises the results given in the first paragraph of the results section. The information in the tables in the supplementary information are additional to the focus of the main text (usually only included to support a specific discussion point) and therefore to avoid cluttering the paper with information that may not be of general interest (for example, the identity of the national parks that recorded different levels of light emissions and skyglow (Table S3) may be of less interest to an international reader than to relevant British or European stakeholders), we have left them in the supplementary information.

REVIEWER: The results' analysis contains obvious observations. For example:

  • "Spearman's rank correlation showed that light emissions were correlated with skyglow and human population (r s = 0.83 and 0.71, respectively), while skyglow was also correlated with human population (r s =0.73)";
  • "Total effects, direct effects and indirect effects all showed greater light emissions from areas containing motorways than from those containing major roads, which in turn were greater than light emissions from areas containing no roads or minor roads (Table 1a; Figure 2e)";
  • "Direct effects for landcover demonstrated that light emissions were least from mountain, heath and bog and were greatest from urban areas (Table 1a; Figure 2b). When considering indirect and total effects light emissions were least in woodland habitat and greatest in urban areas (Table 1a)."

RESPONSE: We have now included the ‘Spearman’s rank correlations’ in Table 1, and have reworded the second and third sentences to avoid information that can easily be gained from examining the tables.

TEXT CHANGED L371-L373: There was a gradient of least light emissions from no roads or minor roads, to most light emissions associated with areas containing motorways, with the total effects from pixels containing motorways being five times greater than from those containing major roads (Table 2a; Figure 2e).

TEXT CHANGED L329-L332: Although direct effects for landcover demonstrated that light emissions were least from mountain, heath and bog, indirect and total effects were least in woodland habitat and all effects were greatest from urban areas (Table 2a; Figure 2b).

REVIEWER: From a technical point of view, the text contains fragments of the template (conflict of interest), and the literature is unordered. It seems to be a draft rather than a final paper submitted to the journal.

RESPONSE: We have edited the ‘conflict of interest’ statement. The numbered citations have been ordered as they first appear in the manuscript.

Conclusions:

REVIEWER: The major part of the paper has to be rewritten.

RESPONSE: We have rewritten the sections that the reviewer has raised as concerns, as detailed above.

REVIEWER: The whole processing path needs a good illustration and clear description. The results should be placed in simple tables with descriptions only in the text.

RESPONSE: Please see our response to this point above.

REVIEWER: The authors should add any justification and comment on the emission model.

RESPONSE: Please see our response to this point above.

REVIEWER: If authors would like to present the technical aspect of calculations (software, commands used, scripts, high-resolution maps), supplementary materials are the right place for it.

RESPONSE: We have now included Figure S2 in the Supplementary information.

REVIEWER: The authors should focus on new and interesting conclusions rather than on evident observations.

RESPONSE: This is the first study to investigate at a resolution of c.500,000 m2 at the national scale the dynamics of where direct emissions and skyglow are produced. We use the most commonly used and sensitive satellite imagery available combined with advanced spatial statistical techniques to determine spatial variation in light emissions and skyglow with human population density and different land uses and structures. Although some of the conclusions can perhaps be considered obvious, this is the first study to provide data on what we may intuitively know. As reviewer 3 stated ‘it is good to actually see the data.’ This study is entirely novel.

REVIEWER: The submission should contain the final paper that is ready to be read by editors, reviewers, and finally - readers.

RESPONSE: With the exception of some text accidentally included with regard to conflicts of interest, the paper follows the guidance to authors provided by the journal.

Recommendation:

Major revision

Reviewer 3 Report

The authors here carryout a much needed full investigation of light pollution in mainland Great Britain. They compare both direct illumination (VIIRS) and skyglow (New World Atlas) across the island to find that although these two types of light pollution are correlated (as we'd expect), that in fact the light pollution in urban areas is spread per capita more so than in rural areas. Although we'd also predict this, it is good to actually see the data. Furthermore, the authors do a great job presenting the data relative to roads and geographic formations. As usual, the group out of Exeter does a great job and although I have studied this manuscript, I don't have any feedback to improve the manuscript. I believe it is ready for publication. There were a few places where I would have changed sentence structure or added a comma, but that was all personal preference, so I have kept those to myself. Nice work!

Author Response

RESPONSE: We are glad that the reviewer enjoyed the manuscript, and we thank them for their encouraging comments.

Round 2

Reviewer 2 Report

The authors did many changes in the paper. These changes improve the quality of it.

Remains a few issues to solve.

The described model is a relation between spatial layers (human, landcover, etc.) and modeled values.
The authors should provide a discussion of combined error: the error of the basic model and SLX model. Otherwise, it suggests that the results of SLX are measured and observed parameters.

Small issues:

Removing the justification of the log-transformation of the data is not the right solution, in my opinion.
I assume that it was done to "normalize" the data. It is enough to check if the distribution of transformed data is normal or not (using any statistical test). Q-Q graf illustrates but not test the distribution normality.
Maybe, another transformation provides a better fitting of the data distribution to normal distribution.

A similar situation is with "approximately normal distributed" error. Statistically, is the error normal or not?

I strongly suggest order literature in any reasonable order. For example, order the list by first author name and a year of publication.

I think that, after solving these issues, the paper is ready to publish.

Author Response

Response to reviewers

REVIEWER: The authors did many changes in the paper. These changes improve the quality of it.

RESPONSE: We are pleased that the reviewer enjoyed the most recent draft of the manuscript.

Remains a few issues to solve.

REVIEWER: The described model is a relation between spatial layers (human, landcover, etc.) and modelled values. The authors should provide a discussion of combined error: the error of the basic model and SLX model. Otherwise, it suggests that the results of SLX are measured and observed parameters.

RESPONSE: We are unclear what the reviewer means by ‘combined error’, and ‘basic model’? By the basic model we suspect the reviewer is referring to the DNB and skyglow layers. The DNB layer is created from composites of measured values the generation of which is described in the methods, while the World Atlas of Artificial Night Sky Brightness layer consists of modelled values from a combination of satellite data and citizen science ground truthing data. However, as is common in analyses of this kind (these are the most commonly used layers for the UK), the ‘predictor’ layers included in the SLX model are also a combination of data that have a marked modelled component (Gridded Human Population Layer (https://sedac.ciesin.columbia.edu/data/collection/gpw-v3 and the UK Landcover Map 2015 (https://www.ceh.ac.uk/services/land-cover-map-2015) - this would be true of all remotely sensed data - and measured data (the road network, and the protected areas layers). Although there will undoubtedly be some degree of error in the peer-reviewed modelled layers, unpicking their inherent error would be a substantial undertaking and lies outside the scope of this study.

Small issues:

REVIEWER: Removing the justification of the log-transformation of the data is not the right solution, in my opinion. 

RESPONSE L276-L277: We have added ‘so that they were approximately normally distributed,’.

REVIEWER: I assume that it was done to "normalize" the data. It is enough to check if the distribution of transformed data is normal or not (using any statistical test). Q-Q graph illustrates but does not test the distribution of normality.

RESPONSE: Following general practice, we believe that normality tests on extremely large samples are not particularly useful. Although parametric tests are usually robust to deviations from normality with large sample sizes (see for example, Lumley et al., 2002), normality tests themselves are extremely sensitive to slight deviations from normality when the sample size is high. For example, the Shapiro test in R has a maximum sample size of 5000, but was originally recommended for use when n < 50 (Shapiro and Wilk, 1965). Our dataset is extremely large (n=45,054). Such large datasets provide sufficient power to detect very minor deviations from normality, and since no dataset of ‘real’ data is truly perfectly normal, these tests are of little value. As is general practice we examined qq-plots visually to check that the data is ‘approximately normally distributed’.

Shapiro, S.S. & Wilk, M.B. (1965) An analysis of variance test for normality (complete samples). Biometrika, 52, 591-611. Doi:10.2307/2333709

Lumley, T., Diehr, P., Emerson, S. & Chen, L. (2002) The importance of the normality assumption in large public health data sets. Annual Review of Public Health, 23, 151-169. http://doi.org/10.1146/annurev.publhealth.23.100901.140546

REVIEWER: Maybe, another transformation provides a better fitting of the data distribution to normal distribution.

RESPONSE: As is normal during the data exploration stage of the analysis we explored a range of transformations on the response variable (please see the accompanying figure). Log transformation was found to make the data closest to normality.

REVIEWER: A similar situation is with "approximately normal distributed" error. Statistically, is the error normal or not?

RESPONSE: Please see our response above. The error distribution is almost certainly not perfectly normal, as is the case for most large real data sets; the question is whether it is sufficiently close to normality for the model result to be robust. This is what we refer to when we state that the error is “approximately normal”.

REVIEWER: I strongly suggest order literature in any reasonable order. For example, order the list by first author name and a year of publication.

RESPONSE: As per the journal’s author guidelines, the literature is numbered in the order it appears in the manuscript. Where more than one paper is cited at the same time, the references are ordered by year, then alphabetically by first author.

REVIEWER: I think that, after solving these issues, the paper is ready to publish.

RESPONSE: We thank the reviewer for their helpful comments through the review process.
